# Optimization of the Treatment of Squamous Cell Carcinoma Cells by Combining Photodynamic Therapy with Cold Atmospheric Plasma

**DOI:** 10.3390/ijms251910808

**Published:** 2024-10-08

**Authors:** Sigrid Karrer, Petra Unger, Nina Spindler, Rolf-Markus Szeimies, Anja Katrin Bosserhoff, Mark Berneburg, Stephanie Arndt

**Affiliations:** 1Department of Dermatology, University Medical Center Regensburg, Franz-Josef-Strauss-Allee 11, 93053 Regensburg, Germany; sigrid.karrer@ukr.de (S.K.); petra.unger@ukr.de (P.U.); nina.spindler@stud.uni-regensburg.de (N.S.); mark.berneburg@ukr.de (M.B.); 2Department of Dermatology and Allergology, Klinikum Vest GmbH Academic Teaching Hospital, 45657 Recklinghausen, Germany; rolf-markus.szeimies@klinikum-vest.de; 3Institute of Biochemistry, Friedrich-Alexander University of Erlangen-Nürnberg, Fahrstr. 17, 91054 Erlangen, Germany; anja.bosserhoff@fau.de

**Keywords:** photodynamic therapy (PDT), 5-aminolaevulinic acid (ALA), cold atmospheric plasma (CAP), actinic keratosis (AK), squamous cell carcinoma (SCC)

## Abstract

Actinic keratosis (AK) is characterized by a reddish or occasionally skin-toned rough patch on sun-damaged skin, and it is regarded as a precursor to squamous cell carcinoma (SCC). Photodynamic therapy (PDT), utilizing 5-aminolevulinic acid (ALA) along with red light, is a recognized treatment option for AK that is limited by the penetration depth of light and the distribution of the photosensitizer into the skin. Cold atmospheric plasma (CAP) is a partially ionized gas with permeability-enhancing and anti-cancer properties. This study analyzed, in vitro, whether a combined treatment of CAP and ALA-PDT may improve the efficacy of the treatment. In addition, the effect of the application sequence of ALA and CAP was investigated using in vitro assays and the molecular characterization of human oral SCC cell lines (SCC-9, SCC-15, SCC-111), human cutaneous SCC cell lines (SCL-1, SCL-2, A431), and normal human epidermal keratinocytes (HEKn). The anti-tumor effect was determined by migration, invasion, and apoptosis assays and supported the improved efficacy of ALA-PDT in combination with CAP. However, the application sequence ALA-CAP–red light seems to be more efficacious than CAP-ALA–red light, which is probably due to increased intracellular ROS levels when ALA is applied first, followed by CAP and red light treatment. Furthermore, the expression of apoptosis- and senescence-related molecules (caspase-3, -6, -9, p16^INK4a^, p21^CIP1^) was increased, and different genes of the junctional network (ZO-1, CX31, CLDN1, CTNNB1) were induced after the combined treatment of CAP plus ALA-PDT. HEKn, however, were much less affected than SCC cells. Overall, the results show that CAP may improve the anti-tumor effects of conventional ALA-PDT on SCC cells. Whether this combined application is successful in treating AK in vivo has to be carefully examined in follow-up studies.

## 1. Introduction

Actinic keratosis (AK) becomes more common with age, especially in the fair-skinned population and on sun-exposed skin [1]. The development of AK is mainly caused by cumulative ultraviolet (UV) exposition, especially to UV-B radiation [1]. UV radiation induces a mutation of the tumor suppressor gene *tp53*, which is considered to be the cause of the development of AK [2]. UV-B radiation leads to transitions from cytidine to thymidine in the tumor suppressor gene *tp53*, resulting in a loss of function of the gene product p53 [2]. As a result, atypical keratinocytes proliferate and migrate in an uncontrolled manner, and AK develops. In most cases, but not necessarily, invasive squamous cell carcinoma (SCC) arises from an AK. Since AK is a potential precursor of SCC, it should be treated [3] (Figure 1).

The choice of therapy for AK should be based on the clinical picture, the risk factors (e.g., immunosuppression, cumulative UV exposure, and the number of lesions), comorbidities, life expectancy and the patient’s wishes [4].

To date, conventional PDT is considered to be one of the best treatment options for multiple AK lesions (field cancerization) [4]. It is non-invasive, has a short sensitization phase, can cover large areas of the skin, and does not affect the surrounding healthy tissue; therefore, conventional PDT leads to impeccable cosmetic results with a short treatment time (single application with possible repeat after three months), while most drug-based procedures take several weeks [4].

Conventional PDT uses the principle of semi-selective, light-induced tissue destruction while maintaining anatomical and physiological integrity. For this purpose, a precursor in the biosynthesis of naturally occurring haem (either 5-aminolevulinic acid (ALA) or methyl aminolaevulinate (MAL)) is administered to the skin tumor for several hours (usually 3 h). During this time, this prodrug penetrates the skin and is preferentially metabolized within tumor cells to protoporphyrin IX (PpIX), the immediate precursor of haem and a very effective photosensitizer. Reactive oxygen species (ROS) are highly reactive molecules that can be generated during PDT when a photosensitizer is activated by light [5]. ROS-induced damage to mitochondria is a critical mechanism through which PDT and other oxidative stress-inducing therapies cause cell death. The release of cytochrome c from damaged mitochondria into the cytoplasm is a key event that triggers the apoptotic pathway.

The initial focus was on the formulation of photosensitizers and the development of suitable illumination devices to improve the effectiveness of therapy and to reduce pain during red light irradiation [6,7,8]. Typically, red light can penetrate the skin to a depth of about 1 to 5 mm and primarily affects the outermost layer of the skin (epidermis). However, current research is focused on the development of physical permeability and penetration enhancing measures. By promoting the absorption and penetration ability of ALA or MAL, PDT may also be used in the future for the curative treatment of more hyperkeratotic AKs and early invasive cutaneous SCC by also improving the PDT response in deep skin layers.

Cold atmospheric plasma (CAP) is often described as an ionized gas that contains charged particles, such as ions and electrons, along with neutral species and various reactive species and photons resulting from the interactions among the plasma components [9]. In tumor research, CAP has shown promising results in numerous in vitro studies, animal models, and clinical case studies [10,11,12,13]. Because tumor cells are more sensitive to ROS and RNS than normal cells, CAP can selectively affect tumor cells [12]. In general, it has emerged that increasing ROS beyond the cancer cell antioxidant protective threshold is an effective strategy for the selective killing of cancer cells [14,15,16,17]. This is because cancer cells maintain a high basal level of ROS due to increased metabolic activity and oncogenic stress [18]. Thus, they are vulnerable to any further increase in ROS. In therapies like PDT or CAP, ROS are used intentionally to exceed the antioxidant protective threshold to damage cancer cells. In addition to this selective anti-tumorigenic property, CAP has the ability to temporarily destabilize the skin barrier. In this way, the transdermal administration of substances (such as ALA) can be improved with a mechanism similar to electroporation [19,20,21,22,23]. There are several strategies and analogs of ALA (Methyl-ALA, Hexyl-ALA, ALA-Esters, N-Ac-ALA and ALA in liposomes or nanoparticles) that have been developed to improve cellular uptake and increase the efficacy of PDT. However, CAP can influence the permeability of cell membranes and combines this ability with anti-tumor properties. This dual function to improve the therapeutic efficacy of PDT is a unique advantage of CAP, which is why it is of importance to clarifying the mechanism.

We hypothesize that both properties of CAP, the release of reactive species and the transient loss of the epidermal barrier to enhance cellular uptake of the prodrug ALA, may have an impact on PDT efficacy.

## 2. Results

The aim of this research work was to examine the use of CAP to increase the effectiveness of conventional ALA-PDT for the treatment of AK. For this investigation and for its future clinical use, the efficacy of combined CAP and ALA-PDT treatment in terms of its anti-tumorigenic behavior was analyzed in vitro using different oral and cutaneous squamous cell carcinoma (SCC) cell lines and human normal keratinocytes (HEKn).

### 2.1. Measurement of Metabolic Activity and Viable Cells Show Cell-Dependent ALA-PDT Treatment Effects

The effectiveness of ALA-PDT depends inter alia on the incubation time and the concentration of the photosensitizer. Depending on these factors, the response of donor cells to ALA-PDT varies greatly. In initial examinations, the individual concentration of ALA was determined for each cell line. Metabolic activity measurements (Figure 2a) together with automated cell count determination (Figure 2b) formed the basis for the ALA concentration determination in the present study. A reduction in metabolic activity of 20–30% was determined using the MTT assay and was indicated as the MA_75_ (metabolic activity of 75%) value in order to be able to detect possible triggering effects of combined CAP and ALA-PDT treatment. Figure 2a shows the dose curves of ALA determined for the different cell lines after an incubation time of 3 h (according to the ALA incubation time commonly used in patients) and illumination with red light. The ALA concentration at which approximately 70–80% metabolic activity is achieved varies greatly between the cell lines used and indicates a donor-dependent response to ALA-PDT. As expected, a high ALA concentration (150 µM) was determined for the non-tumorigenic HEKn cell line. Interestingly, the cutaneous SCL-1 cell line showed an even higher ALA value (220 µM ALA), suggesting a “non-responder” cell line. Table 1 summarizes the optimized ALA concentration for the following cellular and molecular biological investigations. The decrease in metabolic activity observed through the MTT assay does not always indicate how many cells are actually alive (cell viability) [24]. To address this potential discrepancy, we performed additional experiments to count the actual number of cells using a specific automated method called LUNA-FL™, which uses fluorescence to distinguish between living and dead cells. This approach helps provide a more accurate assessment of cell viability alongside the MTT assay results. The measurements were performed 24 h after treatments (CAP, ALA-PDT, CAP-ALA–red light, ALA-CAP–red light) and in untreated control cells, and showed a decrease in the number of viable cells after treatments (Figure 2b).

### 2.2. Elevated DCF Levels as Indicator of Intrinsic ROS in SCC Cells

Reactive oxygen species (ROS) are a group of highly reactive molecules that have been extensively researched in the context of different cancer types. These molecules are generally regarded as normal byproducts of various cellular activities. In most cases, cancer cells display elevated levels of ROS compared to healthy cells, which is attributed to an imbalance between oxidative agents and antioxidant defenses [25]. DCF (2′,7′-dichlorofluorescin) measurement is a widely used method for detecting general ROS activity and is not entirely specific to a single ROS. It reacts with a broad range of ROS, including hydrogen peroxide (H_2_O_2_), hydroxyl radicals (•OH), and peroxynitrite (ONOO−). The comparison of the DCF levels, as an indicator of intracellular ROS, between SCC and HEKn cells clearly showed a ROS induction in all analyzed tumor cells (Figure 3). This characteristic makes SCC cancer cells more sensitive to external stimuli that further increase the production of ROS (e.g., CAP or ALA-PDT).

### 2.3. Combined ALA-CAP–Red Light Treatment Induces Intracellular ROS

Both methods, CAP and ALA-PDT, can effectively induce cell death through ROS production [26,27], but their mechanisms differ. ALA-PDT is more targeted due to the use of a photosensitizer, while CAP affects a broader area, which means that the reactive species produced by CAP can diffuse and interact with a wider range of cells and tissues in the vicinity of the application site rather than being confined to just the cells that have taken up the photosensitizer as in ALA-PDT. However, we were interested in the final ROS levels using the 2′-7′ dichlorofluorescin diacetate (DCFH-DA) assay kit to identify possible triggering effects of the combined treatment. In the study, deliberately low ALA doses (Table 1) were used for the individual cell lines; therefore, the final ROS production was not fully exhausted. This is reflected in the relatively low measured ROS values after ALA-PDT treatment (Figure 4a–g). The ability of CAP to increase cell membrane permeability during treatment, together with production of ROS in large quantities, however, leads to a strong increase in ROS in the CAP-treated cells, except for the SCC-111 cell line (Figure 4f). A ROS quenching effect appears to occur with the CAP-ALA–red light sequence treatment in most of the cell lines, but this has not yet been analyzed further. The relatively low ROS values in the CAP-ALA–red light treatment group could be because the CAP treatment takes place before the 3-h ALA incubation, and the half-life of many of the reactive oxygen species can vary greatly in the range of microseconds to a few milliseconds for superoxide (O_2_-) and from minutes to hours for hydrogen peroxide (H_2_O_2_) for example. In addition, ROS are often very volatile and react quickly with biomolecules (e.g., ALA), which further shortens their half-life. In the ALA-CAP–red light sequence, however, the CAP treatment takes place immediately before the red light irradiation when ALA is already metabolized to protoporphyrin IX (PpIX). Finally, the ROS values from the CAP treatment, together with the ROS production when the photosensitizer is activated by light, lead to a significant increase in ROS, as can be seen in Figure 4 in the ALA-CAP–red light groups. ROS induction was also observed in HEKn cells (Figure 4g). Since the sensitivity to ALA-PDT treatment varies greatly between the analyzed cell lines (Figure 2a,b), we adjusted the ALA doses individually in the present study. When using uniform ALA doses, sensitive cells would be completely killed after ALA-PDT treatment, and trigger effects from the combined treatment with CAP would not be detectable. Since we used ALA efficiency as a benchmark in this study and not ALA quantity, an ALA dose of 150 µM had to be used for HEKn cells in contrast to most of the tumor cells where a dose of 15–90 µM ALA is sufficient (Table 1). For that reason, normal HEKn cells were also affected in the present study. In our initial investigations on dose determination, we observed through cell count measurements that cell viability was only minimally affected when lower doses of ALA (50 µM and 100 µM) were applied to HEKn cells. This supports the hypothesis that the conversion of ALA to PpIX is considerably less in normal cells compared to tumor cells. Consequently, we propose that the rise in ROS levels observed in HEKn cells following ALA-CAP–red light treatment is a result of the elevated ALA dose; however, it does not surpass a critical ROS threshold in these cells.

### 2.4. Cellular and Molecular Effects after Combined CAP and ALA-PDT Treatment

Although ALA-PDT is an important non-invasive therapeutic modality in cancer treatment, it also has adverse side effects, such as the phototoxic damage, apoptosis, or necrosis of healthy cells. Normally, ALA-PDT is designed to protect normal cells as much as possible. However, our aim of the present study was to increase the therapeutic effectiveness on SCC cells using a combined treatment of CAP and ALA-PDT. For this reason, the focus of this study was on ALA efficiency and not on the use of a uniform ALA dose for all cell lines. This leads to a high ALA dose for HEKn normal cells compared to tumor cells (Table 1), which is why certain cytotoxic effects in HEKn cells cannot be avoided under these conditions. We used various cellular and molecular assays (Boyden Chamber migration and invasion assay, FACS Annexin V/PI apoptosis assay, mRNA expression analysis of relevant genes) to analyze the efficacy of a combined CAP and ALA-PDT treatment compared to conventional ALA-PDT. Thereby, the sequence of the treatment (CAP-ALA–red light or ALA-CAP–red light) was also taken into account to see if the effects at the cellular and molecular level are ROS dependent or independent.

#### 2.4.1. Combined CAP and ALA-PDT Treatment Reduces Cell Migration and Invasion of SCC Cells

The directed migration and invasion behavior of cutaneous and oral SCC cells was determined after treatment (CAP, ALA-PDT, CAP-ALA–red light and ALA-CAP–red light) in comparison to untreated cells (ctrl.) by means of Boyden Chamber assays. The directed migration and invasion assays are specific to tumor cells and are not applicable to normal HEKn cells, because normal cells usually do not migrate through the basement membrane (simulated here with an 8 µm pore filter membrane of the Boyden Chamber assay). A 2 min CAP treatment (moderate, non-lethal CAP dose for SCC cells) did not show any significant inhibitory effects on migration (Figure 5a,c) or invasion (Figure 5b,d) in any of the SCC cell lines. ALA-PDT, on the other hand, showed inhibited migration and invasion for all SCC cell lines. This inhibitory effect could be significantly triggered in most SCC cell lines after combined CAP-ALA–red light and ALA-CAP–red light treatment. However, the second combination (ALA-CAP–red light) appeared to better inhibit migration and invasion than the first combination (CAP-ALA–red light).

#### 2.4.2. Combined ALA-CAP–Red Light Treatment Triggers Apoptosis

Using FACS Annexin V/PI analyses 48 h after treatment according to [28,29,30], we could show that apoptosis was not or only marginally induced in SCC cell lines and in HEKn after 2 min CAP treatment in comparison to untreated cells (ctrl.). The 2 min CAP treatment has been shown to be sufficient to induce apoptosis in melanoma cells via FACS Annexin V/PI analyses [31]. For SCC cells, the lethal dose using the plasma care^®^ device (terraplasma GmbH, Garching, Germany) is 5 min. However, to identify potential triggering effects of the combined treatment in this study, we used a non-lethal 2 min CAP dose for SCC cells. Conventional ALA-PDT with the selected ALA doses (Table 1) induced apoptosis only in the SCC-9 cell line (Figure 6a), whereas no increase in apoptosis was observed in the other cell lines (Figure 6b–g). After combined CAP and ALA-PDT treatment, however, apoptosis was significantly increased, particularly after the ALA-CAP–red light sequence. However, apoptosis was also induced in normal HEKn cells. As already mentioned, this effect might be due to the ALA dose of 150 µM used for the treatment of HEKn.

These results were supported at the molecular level by the induction of the mRNA expression of apoptosis- and senescence-related genes (caspase-3, -7, -9, p16^INK4a^, p21^CIP1^) in most of the analyzed SCC cell lines after combined CAP and ALA-PDT treatment in comparison to conventional ALA-PDT (Figure 7). Again, the combination ALA-CAP–red light seems to be more efficacious than the combination CAP-ALA–red light for most SCC cell lines (Figure 7a–f). For HEKn (Figure 7g), a significant increase was only detected for caspase-3 in the treatment sequence ALA-CAP–red light. This finding suggests that normal cells are less sensitive to changes in gene expression after combined treatment than tumor cells.

#### 2.4.3. Combined ALA-CAP–Red Light Treatment Changes Gene Expression of the Junctional Network

In order to detect differences in the permeability behavior of cells after treatment (CAP, ALA-PDT, CAP-ALA–red light, or ALA-CAP–red light), molecular biology studies on the modification of genes of the junctional network (tight junctions (TJs), adherence junctions (AJs) and gap junctions (GJs)) were carried out based on the publication by Schmidt et al., 2020 [32]. The primary focus was on the expression of TJ molecules such as claudin 1 (CLDN1) and zonula occludens 1 (ZO-1), AJ ß-catenin (CTNNB1), and GJ connexin 31 (CX 31) that have already been described to be regulated in vivo in a CAP-dependent manner [32]. The mRNA expression of these selected genes was examined using quantitative real-time PCR analysis. Interestingly, no expression of the junctional genes analyzed was significantly regulated in response to the 2 min CAP treatment in any of the analyzed SCC cell lines (Figure 8a–f), which could be due to the “sublethal” CAP dose used for SCC cell treatment in this study. On the other hand, almost all SCC cell lines (Figure 8a,c–f), including normal HEKn (Figure 8g), showed a noticeable increase in CLDN1, especially after ALA-CAP–red light treatment compared to conventional ALA-PDT. ZO-1, which is responsible for binding integral membrane proteins and anchoring them to the actin cytoskeleton, and was also upregulated after ALA-CAP–red light treatment, although not always significantly, in most of the SCC cell lines (Figure 8a,c,e,f) analyzed, including HEKn (Figure 8g). A similar upregulation was identified for CTNNB1 and CX31 after ALA-CAP–red light treatment in comparison to conventional ALA-PDT in most SCC cell lines. However, these genes were also upregulated in HEKn. An exception was the cutaneous SCL-2 cell line (Figure 8b) and the oral SCC-9 and SCC-111 cell line (Figure 8d,f) that showed no significant regulation of any of these junctional genes after any treatment. These results suggest that these genes of the junctional network also show donor-dependent differences in their regulation after therapeutic treatment.

## 3. Discussion

The inability of topically applied drugs used for PDT (e.g., ALA or MAL) to penetrate deeply enough through keratinized tumor surfaces still represents a major challenge. Cold atmospheric plasma (CAP) has the ability to temporarily destabilize the skin barrier. Such destabilization can facilitate the transdermal administration of substances using a mechanism similar to electroporation [19,20,21,22,23] and may be a way to increase the absorption of ALA into cells. The aim of this study was to use the permeability-enhancing property together with the anti-tumorigenic characteristics of CAP [10,11,12,13] to increase the efficacy of conventional photodynamic therapy (PDT) for the treatment of actinic keratosis (AK). The efficacy of this combined treatment was investigated by analyzing various anti-tumorigenic parameters such as reduced migration and invasion as well as the induction of apoptosis in squamous cell carcinoma (SCC) cell lines of cutaneous (SCL-1, SCL-2 and A431) and oral (SCC-9, SCC-15 and SCC-111) origin. When interpreting the results, the sequence of the treatment (CAP-ALA–red light or ALA-CAP–red light) was also taken into account for the future clinical use of this therapy.

The comparison of oral and cutaneous cell lines did not yield any origin-specific difference in migration or invasion (Figure 5), apoptosis (Figure 6), or gene expression after treatment (Figure 7 and Figure 8). This finding suggests that the origin of the cells has no impact on the success of the treatment per se. We found evidence of this assumption in the study by Dooley et al., 2003, that showed highly similar biomarker profiles between cutaneous and oral SCC cell lines [33].

However, the comparison of the metabolic activity of the analyzed SCC cell lines after ALA-PDT treatment showed significant differences between the donors. The ALA dose, at which approximately 75% of the cells are metabolically active after treatment, varied greatly here (Figure 2a, Table 1). In addition to the decrease in metabolic activity, supplementary cell count determinations 24 h after treatment also revealed reduced viability (Figure 2b). As expected, the non-tumorigenic HEKn cells were least sensitive to treatment, followed by the cutaneous SCL-1 cell line. Here, we suspected a “non-responder” cell line. However, the functional migration-, invasion-, and apoptosis examination (Figure 5 and Figure 6) showed a clear response of SCL-1 cells to both ALA-PDT treatments and an even better anti-tumorigenic effect after combined CAP and ALA-PDT treatment. These results clearly show that the therapeutic response is donor-dependent and that a possible adjustment of the parameters (e.g., ALA dose, ALA incubation time, and red light treatment) should be considered if there is no response to therapy.

In addition to therapeutic improvements by combining CAP and ALA-PDT, the aim was to protect normal cells (HEKn) as much as possible during treatment. Since the focus in the present study was on ALA efficiency and not on the use of a uniform ALA dose for all cell lines, damage to normal HEKn cells was not erroneous in the present study. This is due to the necessary ALA dose of 150 µM for HEKn cells compared to the doses used in tumor cells (Table 1) to achieve the same metabolic activity value of 75% (MA_75_). If HEKn cells were treated with lower ALA doses as established for most SCC cell lines (15–90 µM), they would most likely be less affected or not be affected at all. However, this must be analyzed in detail in follow-up examinations. Since the intrinsic ROS value in HEKn cells is lower than that in all SCC cell lines analyzed (Figure 3), we assume that the lethal ROS threshold in HEKn cells is not exceeded under treatment conditions normally used for SCC cell lines. However, detailed follow-up studies must clarify how the combined treatment of CAP and ALA-PDT affects tumor cells and normal cells in vivo and what ALA doses are required to treat AK in a tumor model.

The anti-cancer effects of CAP on AK and SCC cells have already been reported in various preclinical studies and in small clinical case studies [34,35,36,37]. CAP treatment showed a selective growth-reducing effect on SCC cells in vitro [37] and was able to stop the progression of UV-B-induced SCC-like skin lesions without affecting the healthy skin in vivo [34]. However, the biological mode of action of the anti-tumor potential of CAP is not yet fully understood, which is why no larger clinical studies with CAP have yet been carried out with tumor patients. Effects on the cell cycle, senescence, apoptosis, and necrosis caused by CAP-generated reactive oxygen and nitrogen species (RONS) have been described as causal [38]. In the present study, we chose a moderate, non-lethal CAP dose (2 min; 4 kHz) for all analyzed SCC cell lines to detect potential triggering effects when using the combined treatment; thus, we observed hardly any anti-tumorigenic effects after a single 2 min CAP treatment (Figure 5, Figure 6 and Figure 7). However, when using the combined treatment, the anti-tumorigenic effects were strongly improved, primarily after the ALA-CAP–red light sequence. This result was a surprise to us, as we initially focused on the permeability-promoting property of CAP to improve the absorption of ALA into tumor cells and, for future in vivo applications, to promote the penetration of protoporphyrin IX (PpIX) into deeper cell layers. For this purpose, the sequence CAP-ALA–red light was chosen, but it seems to be less efficacious. The ALA-CAP–red light sequence was used to accumulate the ROS values of CAP and PDT. Because this combination was far more efficacious in most of the SCC cell lines analyzed, we speculate that the lethal ROS threshold for tumor cells was exceeded when using this sequence. These results suggest that the efficacy of PDT could be improved by a combination with CAP. To prove this assumption, the efficacy of CAP-ALA–red light and ALA-CAP–red light should be investigated in a clinical trial in patients with AK and compared to ALA-PDT and CAP alone. As a practical approach, an additional application of CAP would be simple and fast to perform; contraindications are not known yet. The efficacy of CAP in the treatment of AK has already been demonstrated in some case series [35,39], but due to insufficient studies, this application is not yet included in the S2k guideline “Rational therapeutic use of cold physical plasma” [40]. However, the management of CAP application for the treatment of AK is similar to the application of CAP for the treatment of wounds and takes place either before ALA treatment or immediately afterward. Several methods exist for delivering CAP to the skin for the treatment of AK, such as Plasma Jet Devices, Dielectric Barrier Discharge (DBD), and Plasma Pens. In our study, we used a prototype of the plasma care^®^ device, an advanced version of surface micro-discharge technology (SMD). While each of these devices functions based on different principles for generating CAP, they all allow for direct application to the skin’s surface. Clinical studies are necessary to determine whether a single 2-min CAP treatment, as utilized in our cell culture study, is adequate when combined with PDT. The question if the side effects of PDT (e.g., pain during illumination) are modified or intensified by adding CAP should also be addressed in a clinical study. However, it cannot be assumed that the combined treatment is equally effective for all patients. There are also so-called ALA-PDT non-responders. Thus, the combined treatment does not guarantee an increase in effectiveness for all patients. Since it is not yet known why some patients do not respond to ALA-PDT treatment, it is difficult to speculate how the limits of the combined therapy can be assessed. Another aspect that needs to be clarified is the effect of the reactive species produced during both ALA-PDT and CAP treatment. Follow-up in vivo studies will have to show whether the combination of both therapies is likely to result in increased damage to the normal surrounding tissue. If a combination treatment regimen proves safe and effective for AK, other indications for PDT like Bowen’s disease or superficial basal cell carcinoma might profit from this approach.

Finally, we investigated the influence of CAP, ALA-PDT, and the combined treatments on the junctional network in SCC cells and normal HEKs (Figure 8), because the study by Schmidt et al., 2020, had already described CAP-induced changes in the expression of genes involved in maintaining the barrier function [32]. Selected transcript codings for proteins of cell–cell connections in TJs (CLDN1, ZO-1), AJs (CTNNB1), and GJs (CX31) were described as upregulated in intact murine skin after CAP treatment (3 s, 20 s; 1 MHz; kINPen Med, neoplas tools, Greifswald, Germany). In our study using different SCC cell lines, these genes were not uniformly induced in the analyzed SCC cell lines or in HEKn cells after CAP treatment (2 min; 4 kHz). The average induction after CAP treatment was about 2–3-fold. However, many of these genes were significantly induced after the combined treatment. The best effects were again observed after treatment with the ALA-CAP–red light sequence. The question of how CAP in combination with ALA-PDT regulates these genes and how it influences the barrier function of cells needs to be analyzed in future studies.

Even though this in vitro study is a good way of investigating cell and molecular biological effects following combined CAP + ALA-PDT treatments, this study had limitations that should not be ignored. A key limitation of this study is that it was conducted in vitro, meaning that the findings may not be directly applicable to patient situations. To establish the efficacy and safety of this combined therapy, clinical studies are necessary. Additionally, the experiments were performed using monolayer and monoculture systems, which further restricts the applicability of the results. The depth of penetration of the prodrug ALA is critical for the success of the treatment in vivo. Future research using ex vivo skin models is needed to determine if a combined treatment can enhance ALA’s penetration depth. Moreover, in vivo studies highlight the importance of paracrine mechanisms, where cells affected by the therapy communicate with other cell types in deeper layers that are not directly treated. It remains to be investigated whether paracrine signaling is involved in the combined treatment, which could be explored through co-culture experiments or in vivo studies. Of course, the study design, which utilized ALA efficiency as a benchmark without employing a uniform ALA dose, has limits. The treatment effects cannot be directly compared across the different cell lines, and the high ALA dose necessary for normal cells inevitably lead to cell damage in HEKn cells, which should be avoided in vivo. Nevertheless, the efficacy-based study design shows us synergistic effects when ALA-PDT is combined with CAP, which is why the limitations listed are accepted in the present in vitro study.

## 4. Materials and Methods

### 4.1. Plasma Device

A prototype of the plasma care^®^ device, developed by terraplasma GmbH in Garching, Germany, was utilized for cold atmospheric plasma (CAP) treatment. A recent publication detailed the device’s design, technology, and ozone emission spectrum [41]. This prototype enables frequency adjustments between an oxygen mode (4 kHz) and a nitrogen mode (8 kHz), with this study specifically utilizing the 4 kHz setting. The device employs a technology known as “thin-film technology”, which is an advanced version of surface micro-discharge technology (SMD) [31]. The application of a high voltage of 3.5 kV generates millimeter-sized micro-discharges within the plasma source unit. This unit is composed of a high-voltage electrode, a dielectric, and a grounded structured electrode [41], which together produce plasma components that can be adjusted based on frequency and voltage. To ensure a consistent distance between the device and the Petri dish (35 mm, Corning, Merck, Darmstadt, Germany) containing the cultured cells, the device was positioned on a spacer [41], creating an isolated treatment area.

### 4.2. Cell Lines and Cell Culture Conditions

In the study, 3 oral and 3 cutaneous SCC cell lines were used. HEKn were used as non-tumorigenic cells. The oral squamous cell carcinoma cell lines SCC-15 (ATCC^®^CRL-1623™) and SCC-9 (ATCC^®^CRL-1629™) and the cutaneous squamous cell carcinoma cell line A-431 (ATCC^®^CRL-1555™) were purchased from ATCC (Manassa, VA 20110-2209, USA), and the oral SCC cell line SCC-111 was obtained from DSMZ (DSMZ-German Collection of Microorganisms and Cell Cultures GmbH, Braunschweig, Germany). Both cutaneous SCC cell lines SCL-1 and SCL-2 were purchased from Cell Lines Service (CLS, Hölzel Diagnostika Handels GmbH, Köln, Germany). Human epidermal keratinocytes, neonatal (HEKn) were ordered via CellSystems GmbH (Troisdorf, Germany). All SCC cell lines, with the exception of SCC-111, were cultured in Dulbecco’s Modified Eagle’s Medium (DMEM) (ThermoFisher Scientific, Schwerte, Germany), which was supplemented with 10% fetal bovine serum (FBS; Anprotec, Bruckberg, Germany), 1% penicillin/streptomycin (P/S; Sigma Aldrich GmbH, Steinheim, Germany), 1% L-glutamine (Sigma Aldrich GmbH, Steinheim, Germany), and 1% Amphotericin B (Sigma Aldrich GmbH). In contrast, SCC-111 cells were grown in DMEM containing 20% FBS and supplemented with 1% Minimum Essential Medium Non-Essential Amino Acids (MEM NEAA (100×); ThermoFisher Scientific). HEKn were cultivated in a DermaLife^®^ K Medium Complete Kit (DermaLife Basal Medium (LM-0004), DermaLife K LifeFactors^®^Kit (LS-1030) from CellSystems GmbH (Troisdorf, Germany)). All cells were maintained in 75 cm^2^ Falcon^®^ cell culture flasks (Corning Merck, Darmstadt, Germany) and incubated under humidified conditions in a 5% CO_2_ incubator (Life Technologies GmbH, Darmstadt, Germany) at 37 °C. The cells were passaged at a ratio of 1:3 every three days. Following a wash with Dulbecco’s Phosphate-Buffered Saline (DPBS) (ThermoFisher Scientific), a solution of 0.05% trypsin/0.02% EDTA (Sigma Aldrich GmbH) in DPBS was used to detach the cells. After centrifugation and the removal of the trypsin solution, the cells were resuspended and counted using Luna FL Acridine Orange (AO)/Propidium Iodide (PI) Stain (BioCat GmbH, Heidelberg, Germany). Mycoplasma contamination was routinely checked and excluded according to the manufacturer’s guidelines provided with the PCR Mycoplasma Test Kit (PanReac AppliChem, Darmstadt, Germany).

### 4.3. Treatment of Cells with Cold Atmospheric Plasma (CAP)

For treatment, 250,000 cells were seeded into 35 mm Petri dishes (Corning, Merck, Darmstadt, Germany) and incubated for 24 h. Immediately before CAP treatment, the cell culture medium was removed, and the cells were covered with 250 µL DPBS (ThermoFisher Scientific). CAP treatment in DPBS was chosen because ROS quenching effects were observed when using medium with fetal bovine serum (FBS) during treatment. This liquid film is necessary to protect the cells against dehydration during CAP treatment. To guarantee a standardized distance between the device and the Petri dish (35 mm, Corning, Merck, Darmstadt, Germany), the device was placed onto a spacer. All SCC cells and HEKn cells were exposed to CAP (2 min; 4 kHz). In contrast to melanoma cells, in which the lethal dose of CAP is already reached after 2 min [31], the lethal dose for SCC cells is only reached after 5 min of CAP treatment. In this study, however, a “sublethal” CAP dose was deliberately used in order to be able to adequately detect potential triggering effects when using the combined treatments.

### 4.4. Treatment of Cells with 5-Aminolaevulinic Acid (ALA)

Immediately after CAP treatment, DPBS was replaced with 2.5 mL serum-free cell culture medium containing 5-Aminolaevulinic Acid (5-ALA; short ALA) (15–220 µM; Merck AG, Darmstadt, Germany), and cells were allowed to take up ALA for 3 h. We used serum-free medium for cell culture treatments because the uptake of ALA can be influenced by the presence of fetal bovine serum (FBS) in the cell culture medium. FBS contains a variety of proteins, including albumin and other serum proteins, which can bind to ALA. This binding can potentially reduce the free concentration of ALA available in the culture medium and affect its uptake by cells. In addition, serum proteins might compete with ALA for cellular uptake or for binding sites on cell membranes. This could influence the efficiency of ALA entering the cells. Furthermore, the presence of serum can change the osmotic balance and ion concentrations in the medium, which might affect the cell membrane’s permeability and, consequently, the uptake of ALA.

The ALA concentration was determined individually for each cell line using metabolic activity analyzes (see Section 4.6) supported with cell count determinations (see Section 4.6). A moderate reduction in metabolic activity of 20–30% (MA_75_) was chosen to be able to recognize potential triggering effects when using combined treatments with CAP. Table 1 gives an overview of the established ALA concentration for the individual cell lines. After ALA incubation, the medium containing ALA was removed, and the cells were rinsed and submerged in 2.5 mL DPBS for red light irradiation (see Section 4.5).

### 4.5. Red Light Irradiation Parameter and Light Source

The cell culture was irradiated with red light using an incoherent light source (λ_em_ = 575–750 nm, PDT-1200L, Waldmann Medizintechnik, Schwenningen, Germany) with a light intensity of 160 mW/cm^2^ (100 J/cm^2^). For conventional ALA-PDT treatment, red light treatment was carried out immediately after ALA incubation for 3 h. For this purpose, the cell culture medium containing ALA was removed and the cells were washed with DPBS. The red light treatment was carried out with fresh 2.5 mL DPBS. Immediately after the treatment, the DPBS was replaced again with the appropriate cell culture medium, and the cells were incubated for 24 or 48 h. In the combined CAP-ALA–red light application, the cells were treated with CAP for 2 min (see Section 4.3). Immediately after CAP treatment, the cells were incubated in ALA for 3 h (see Section 4.4) and treated with red light directly after in analogy to conventional ALA-PDT with 2.5 mL DPBS. In the ALA-CAP–red light application, cells were incubated in ALA for 3 h and treated with CAP for 2 min. Immediately after CAP treatment, cells were treated with red light.

### 4.6. Measurement of Metabolic Activity and Cell Count Determination

Metabolic activity was determined with a 3-[4,5-dimethylthiazol-2-yl]-2,5-diphenyl-tetrazolium bromide (MTT) test. The MTT assay is used to measure cellular metabolic activity as an indicator of cell viability, proliferation, and cytotoxicity [42,43]. To confirm that metabolic activity correlates with viability, additional cell count measurements were performed. To assess cell viability and exclude non-viable cells, the untreated and treated cells (CAP, ALA-PDT, CAP-ALA–red light, ALA-CAP–red light) were stained with Acridine Orange (AO)/Propidium Iodide (PI) 24 h after treatment and analyzed using LUNA-FL™ in an automated fluorescence cell counting mode according to the manufacturer’s instructions (Logos Biosystems, Villeneuve d’Ascq, France).

To evaluate metabolic activity after treatment with different concentrations of ALA (0–250 µM), 10^4^ cells/well were seeded onto a 96-well microtiter plate (Corning, Merck, Darmstadt, Germany). For ALA incubation (3 h) (see Section 4.4), cells were maintained in 100 µL of medium without FBS. FBS provides nutrients and growth factors that might alter cellular metabolism and the cellular uptake of various compounds, including ALA. In experimental setups where precise control over ALA uptake is necessary, it is therefore useful to use serum-free medium during incubation. After ALA incubation, the medium was replaced by 100 µL DPBS, and the microtiter plate was irradiated with red light (see Section 4.5). DPBS was exchanged with cell culture medium, and cells were incubated under humid conditions in a 5% CO_2_ incubator at 37 °C for 24 h. In total, 10 µL of MTT reagent (5 mg/mL in DPBS, Sigma-Aldrich) was added directly to the medium in the wells of the microtiter plate and incubated for 4 h. Insoluble formazan was formed in the cells in proportion to the activity of the dehydrogenases. The formazan was finally dissolved in 100 µL 20% SDS (sodium dodecyl sulfate, Sigma-Aldrich) and incubated overnight. The absorbance of the dissolved solution was observed with a microtiter plate reader (MWG-Biotech) at 540 nm. Each sample was assayed in duplicate and the entire experiments were conducted three times.

### 4.7. Measurement of Intracellular Reactive Oxygen Species (ROS)

For the measurement of intracellular ROS, 10^4^ cells/well were seeded onto a black 96-well microtiter plate (PerkinElmer, VWR, Ismaning, Germany). Then, 24 h after seeding the cells, treatments with CAP (4 kHz; 2 min) and ALA (3 h) were carried out, and the cell permeable reagent 2′-7′dichlorofluorescin diacetate (DCFH-DA) was added and incubated for 60 min at the cells’ optimal temperature in dark conditions immediately after the CAP treatment (Section 4.3) or after a 2 h incubation period with ALA (Section 4.4). Subsequently, treatment with red light (Section 4.5) was carried out in the corresponding approaches (ALA-PDT, CAP-ALA–red light, ALA-CAP–red light). DCFH-DA is a fluorogenic dye used to assess the activity of hydroxyl, peroxyl, and other reactive oxygen species (ROS). Once taken up by cells, DCFH-DA is deacetylated by cellular esterases into a non-fluorescent compound that was subsequently oxidized by ROS to form 2’-7’dichlorofluorescein (DCF). DCF is a fluorescent compound that can be quantified using a plate reader (Varioscan Flash, Thermo Fisher, Schwerte, Germany), with a maximum excitation wavelength of 495 nm and a maximum emission wavelength of 529 nm. Measurements were taken 60 min after the addition of DCFH-DA to the samples. The Assay Kit (DCFH-DA Redox Probe; G-Biosciences, 9800 Page Avenue St. Louis, MO 63132-1429, USA) was used as specified by the manufacturer.

### 4.8. Migration and Invasion of Cells

Migration and invasion assays were performed using Boyden Chambers containing cell culture inserts with 8 µm pore filter membranes (Corning, Kaiserslautern, Germany) placed onto 24-well cell culture plates (Corning, Kaiserslautern Germany). The filter of a culture insert remained untreated (migration) or was coated with 30 µL matrigel (diluted 1:3 in H_2_O; Becton Dickinson, Heidelberg, Germany) for invasion analysis. Matrigel was evenly distributed without air bubbles before hardening in an incubator for 30 min. The lower compartment of the chamber system was then filled with 500 µL fibroblast-conditioned medium, used as a chemo-attractant. In total, 150,000 SCC cells (migration) and 100,000 SCC cells (invasion) were then placed in 500 µL FBS-free cell culture medium into the cell culture insert and were incubated under humid atmosphere (5% CO_2_, 37 °C) for 4 h (migration: SCC-9, SCC-15, SCC-111) and for 18 h (migration: SCL-1, SCL-2, A431). Hence, invasion assays were incubated for 18 h (invasion: SCL-1, SCL-2, A431, SCC-15, SCC-111) and for 4 h (invasion: SCC-9). After incubation for the indicated time, the matrigel-coated inserts were cleaned with DPBS-soaked cotton swabs, fixed, and stained with the Hemacolor^®^ staining kit (Merck KGaA, Darmstadt, Germany) according to the manufacturer’s instructions. The filters of the inserts were cut out with a scalpel, placed upside down on a microscope slide (ThermoFisher Scientific), and covered with Neo-Mount™ (Sigma-Aldrich). Three filters were stained per experiment and treated. Five fields of view were photographed and evaluated for each filter. All migrated and invaded cells were photographed and counted semi-automatically using the All-in-One Fluorescence Microscope BZ-X800 (Keyence, Neu-Isenburg, Germany) and the cell counting and analysis software BZ-H4C (Keyence, Neu-Isenburg, Germany). All automated cell counts were checked manually and any missing cells were added manually. To improve comparability of the individual cell lines, the respective untreated controls (ctrl.) were finally set to 100%, and the treated groups were set in relation to this percentage. All assays were repeated at least three times.

### 4.9. Measurement of Cell Apoptosis and Necrosis

For analyzing apoptosis and necrosis, 250,000 cells in 35 mm Petri dishes were treated with CAP, ALA-PDT, CAP-ALA–red light, or ALA-CAP–red-light or remained untreated as described above (Section 4.3, Section 4.4 and Section 4.5). Apoptotic cells were investigated by flow cytometry 48 h after treatment using the FITC Annexin V Apoptosis Detection Kit with Propidium Iodide (PI) (BioLegend, Koblenz, Germany) according to the manufacturer’s instructions and published elsewhere [28]. Flow cytometry analysis was carried out with a FACS Calibur Flow Cytometer (Becton Dickinson, Heidelberg, Germany). FACS data were analyzed using the FlowJo™ v10 software. Experiments were conducted in duplicate and repeated three times. The measurement time of 48 h after treatment was chosen carefully, because the induction of apoptosis physiologically takes time, as comparative studies have shown [28,29,30,44]. In addition, the parameters of CAP dose (2 min; 4 kHz) and ALA dose (Table 1) were relatively low in order to still be able to detect trigger effects when using the combined treatments. So, we did not expect strong apoptotic effects directly after treatment.

### 4.10. Isolation of Ribonucleic Acid (RNA) and Reverse Transcription

RNA from 250,000 untreated (ctrl.) or treated (CAP, ALA-PDT, CAP-ALA–red light, or ALA-CAP–red light) cells was isolated 24 h after treatment using the NucleoSpin^®^ RNA Plus Kit (Macherey-Nagel, Düren, Germany) according to the manufacturer’s instructions. cDNA was generated with the SuperScript II Reverse Transcriptase kit (Invitrogen; Thermo Fisher Scientific, Darmstadt, Germany) using 2–5 µg of total RNA for transcription according to the manufacturer’s instructions.

### 4.11. Quantitative Real-Time Polymerase Chain Reaction (PCR) Analysis

Gene expression analysis was conducted using quantitative real-time PCR with specific primer sets (Sigma-Aldrich, Steinheim, Germany) and conditions outlined in Table 2. The analysis was carried out utilizing LightCycler technology (Roche Diagnostics, Mannheim, Germany), following the procedures described in previous studies [45]. PCRs were assessed through melting curve analysis. To confirm the integrity of cDNA and to normalize expression levels, beta-actin (β-actin) was amplified. Each experiment was conducted a minimum of three times, with each run performed in duplicate.

### 4.12. Statistical Analysis

All data were analyzed using GraphPad Prism 10.2.2 software (GraphPad Software Inc., San Diego, CA, USA) and presented as mean values ± standard deviation (SD). To determine significant differences among the various groups, ordinary one-way ANOVA was performed, followed by Bonferroni’s or Tukey’s multiple comparison tests. Significant results are indicated at * *p* ≤ 0.05, ** *p* < 0.01, *** *p* < 0.001, or **** *p* < 0.0001. A detailed description of the statistics is given in the corresponding legend below the respective figure.

## 5. Conclusions

Cold atmospheric plasma (CAP) is a rapidly growing new research area in health care. One promising novel medical application of CAP is the treatment of actinic keratosis (AK). Even though research into the treatment of tumor diseases with CAP has not yet progressed that far, it offers a wide range of new therapeutic options for various tumor diseases in the future. The permeability-promoting property together with the anti-tumorigenic effect of CAP may improve the effectiveness of conventional therapies such as photodynamic therapy (PDT) with 5-aminolevulinic acid (ALA). This preclinical study on various oral and cutaneous squamous cell carcinoma (SCC) cell lines showed an increase in anti-tumor effects when combining CAP with ALA-PDT. The sequence of the application of CAP and ALA prior to red light illumination (either CAP before ALA incubation for 3 h or CAP after 3 h of ALA incubation) will contribute significantly to the success of the therapy. Here, the ALA-CAP–red light sequence achieved much better results, most likely due to the exceeded lethal reactive oxygen species (ROS) threshold value for tumor cells. The extent to which this combined treatment promotes the therapeutic effect in vivo needs to be carefully examined in follow-up studies.

## Figures and Tables

**Figure 1 ijms-25-10808-f001:**
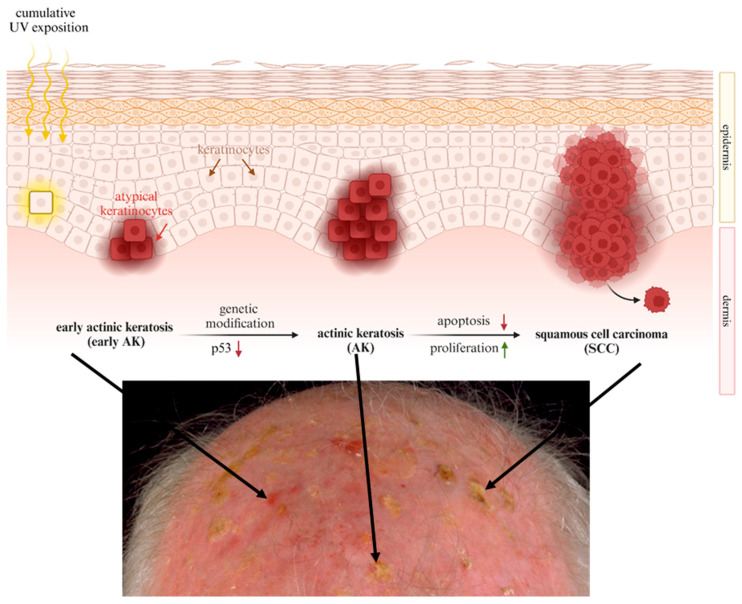
Onset and progression of squamous cell carcinoma (SCC). Early actinic keratosis (AK) is mainly caused by chronic ultraviolet (UV) light exposure. UV radiation induces a mutation of the tumor suppressor gene *tp53*, which is considered to be one cause of the development of AK. As a result, atypical keratinocytes proliferate in an uncontrolled manner, and apoptosis is reduced. AK is regarded as a potential precursor of squamous cell carcinoma (SCC). (red arrow: reduction; green arrow: induction). Created with BioRender.com.

**Figure 2 ijms-25-10808-f002:**
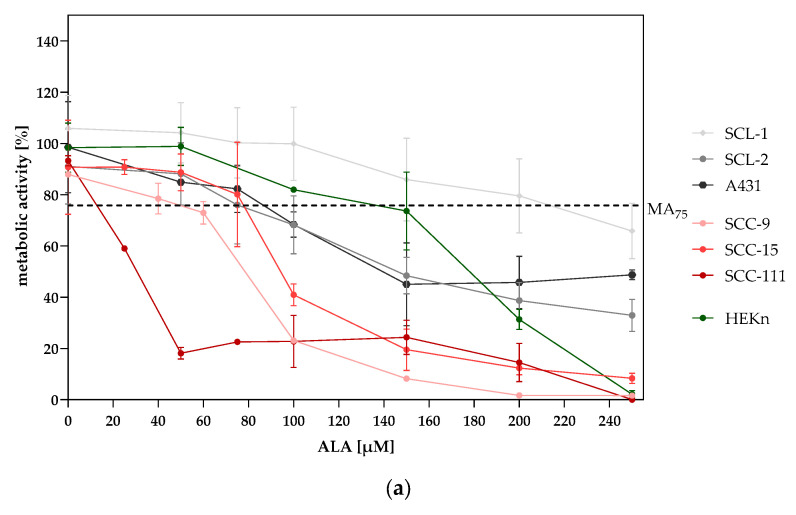
(**a**) Metabolic activity curves of cutaneous SCC cell lines (SCL-1, SCL-2, A431), oral SCC cell lines (SCC-9, SCC-15, SCC-111), and HEKn cells were generated 24 h after ALA incubation for 3 h followed by red light treatment (100 J/cm^2^; 160 mW/cm^2^). The ALA concentration at which approximately 70–80% of cells are metabolically active was defined as the MA_75_ value and served as the individual ALA concentration for cell culture examinations. (**b**) The numbers of living cells of (a–c) cutaneous SCC cells (SCL-1, SCL-2, A431), (d–f) oral SCC cells (SCC-9, SCC-15, SCC-111), and (g) HEKn cells were determined 24 h after treatments (CAP, ALA-PDT, CAP-ALA–red light, ALA-CAP–red light) and in untreated cells (ctrl.) using Acridine Orange (AO)/Propidium Iodide (PI) staining and were analyzed using LUNA-FL™ in an automated fluorescence cell counting mode The results are the means of a single experiment performed in duplicate.

**Figure 3 ijms-25-10808-f003:**
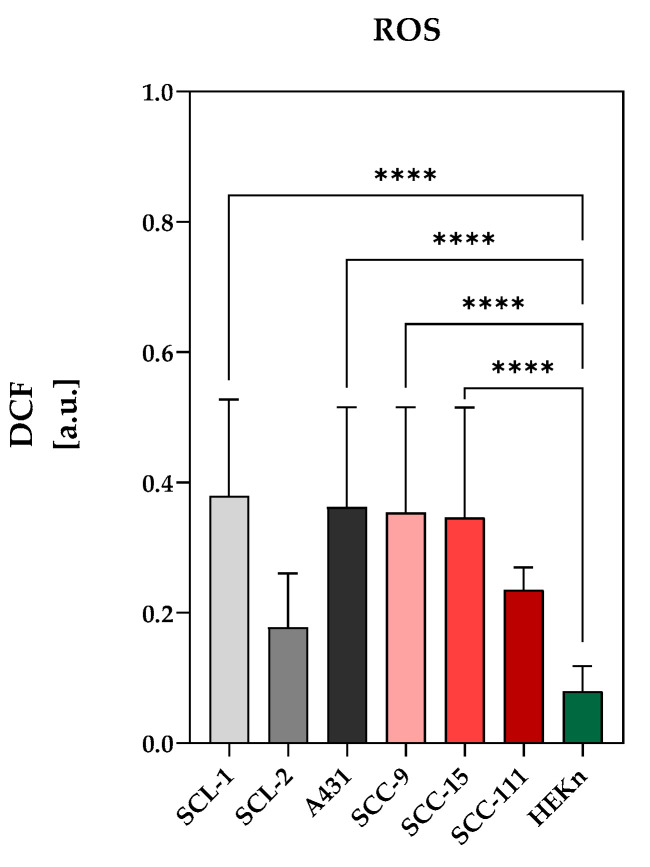
Determination of DCF in arbitrary units [a.u.] as indicator of intracellular ROS levels in untreated SCC and HEKn cells using fluorogenic DCFH-DA assay. Intracellular ROS levels in cutaneous SCC cells (SCL-1, SCL-2, A431) and oral SCC cells (SCC-9, SCC-15, SCC-11) in comparison to non-tumorigenic HEKn cells. The results are the means of three independent measurements. Statistical analysis: Ordinary one-way ANOVA with Tukey’s multiple comparison test was carried out to compare the means of HEKn and SCC cell lines. **** *p* < 0.0001.

**Figure 4 ijms-25-10808-f004:**
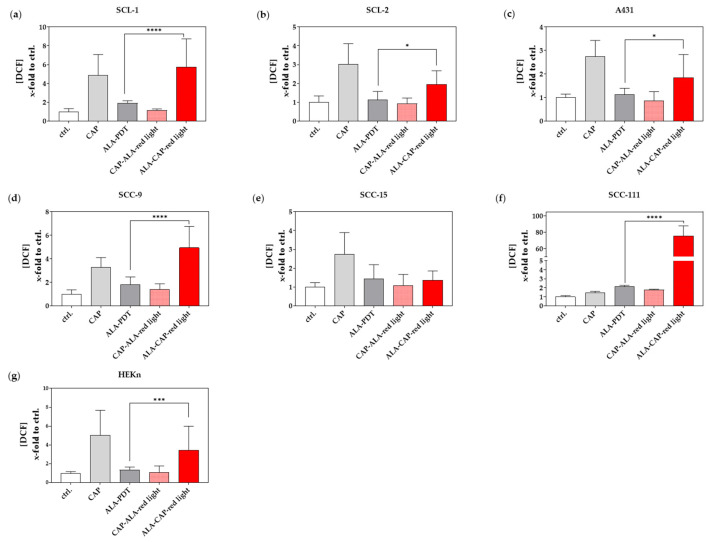
The determination of DCF as an indicator of intracellular ROS levels in treated SCC and HEKn cells using a fluorogenic DCFH-DA assay. DCF (x-fold to ctrl.) was determined in cutaneous SCC cell lines (**a**) SCL-1, (**b**) SCL-2, and (**c**) A431; in oral SCC cell lines (**d**) SCC-9, (**e**) SCC-15 and (**f**) SCC-111; and in (**g**) normal HEKn after CAP treatment, ALA-PDT and after combined treatments (CAP-ALA–red light, ALA-CAP–red light). Statistical analysis: Ordinary one-way ANOVA with Bonferroni’s multiple comparison test was carried out to compare the mean of ALA-PDT with the results of the combined CAP-ALA–red light and ALA-CAP–red light treatment. * *p* ≤ 0.05, *** *p* < 0.001, **** *p* < 0.0001.

**Figure 5 ijms-25-10808-f005:**
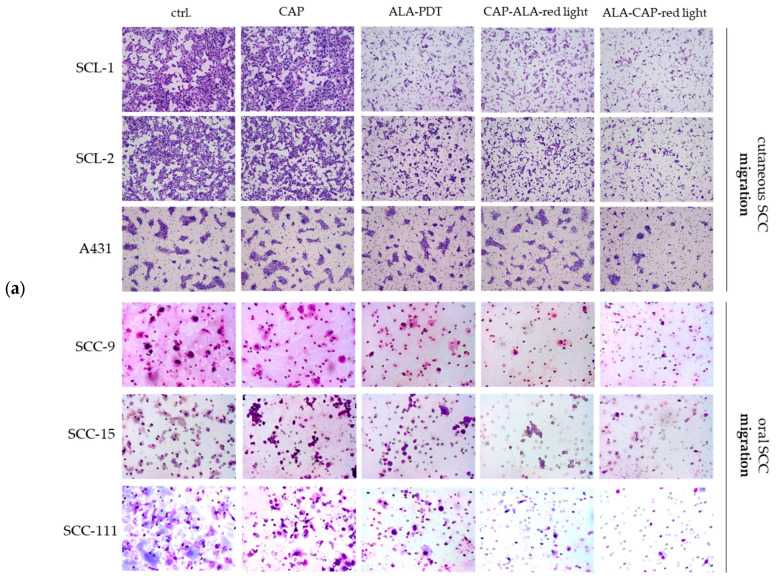
Migration and invasion of SCC cells after conventional ALA-PDT and after combined treatments (CAP-ALA–red light, ALA-CAP–red light). (**a**) Boyden Chamber migration assay. Exemplary overview of stained 8 µm pore filter membranes of migrated cutaneous SCC cells (SCL-1, SCL-2, A431) and oral SCC cells (SCC-9, SCC-15, SCC-111). (**b**) Boyden Chamber invasion assay. Exemplary overview of stained 8 µm pore filter membranes of matrigel-invaded cutaneous SCC cells (SCL-1, SCL-2, A431) and oral SCC cells (SCC-9, SCC-15, SCC-111). Quantification and statistical examination of (**c**) migrated and (**d**) invaded SCC cells. Each experiment was carried out in triplicates. Of each experiment, at least five representative images per group were taken at 20-fold magnification, and the cells per field of view were counted and summarized. Statistical analysis: Ordinary one-way ANOVA with Bonferroni’s multiple comparison test was used with * *p* ≤ 0.05, ** *p* < 0.01, *** *p* < 0.001 and **** *p* < 0.0001 to indicate the mean differences within the conventional ALA-PDT and the combined treatments (CAP-ALA–red light, ALA-CAP–red light).

**Figure 6 ijms-25-10808-f006:**
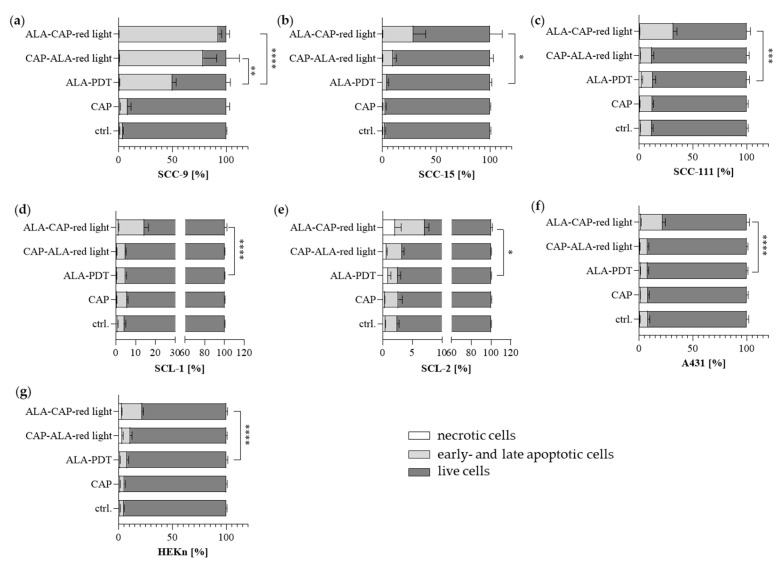
Annexin V/PI double-staining assay was performed 48 h after different treatments (CAP, ALA-PDT, CAP-ALA–red light, and ALA-CAP–red light) and compared to untreated control cells (ctrl.). Apoptosis (early and late apoptosis), necrosis, and the number of live cells were analyzed in oral SCC cell lines (**a**) SCC-9, (**b**) SCC-15 and (**c**) SCC-111; cutaneous SCC cell lines (**d**) SCL-1, (**e**) SCL-2 and (**f**) A431; and (**g**) HEKn cells. The graphs present the percentage (mean ± SD) of the cells in the region among the total cells from three independent experiments in duplicate. Statistical analysis: Ordinary one-way ANOVA with Bonferroni’s multiple comparison test was used with * *p* ≤ 0.05, ** *p* < 0.01, *** *p* < 0.001 and **** *p* < 0.0001 to indicate the mean differences within the conventional ALA-PDT and the combined treatment groups (CAP-ALA–red light and ALA-CAP–red light).

**Figure 7 ijms-25-10808-f007:**
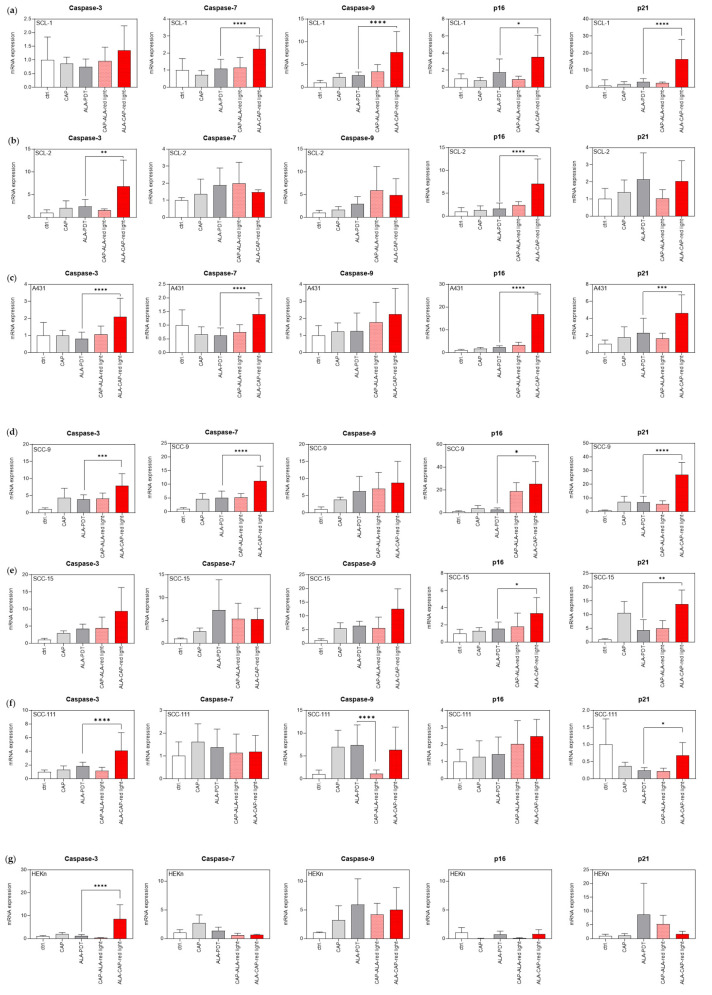
Expression of apoptosis- and senescence-related molecules on the mRNA level after different treatments (CAP, ALA-PDT, CAP-ALA–red light and ALA-CAP–red light) and in untreated cells (ctrl.). (**a**–**c**) mRNA expression (caspase-3, -7, -9, p16^INK4a^, p21^CIP1^) in cutaneous SCC cell lines (SCL-1, SCL-2, A431), (**d**–**f**) in oral SCC cell lines, and (**g**) HEKn was measured 24 h after treatment. Statistical analysis: Ordinary one-way ANOVA with Bonferroni’s multiple comparison test was carried out to compare the mean of conventional ALA-PDT with combined treatments (CAP-ALA–red light, ALA–CAP-red light). * *p* ≤ 0.05, ** *p* < 0.01, *** *p* < 0.001, **** *p* < 0.0001.

**Figure 8 ijms-25-10808-f008:**
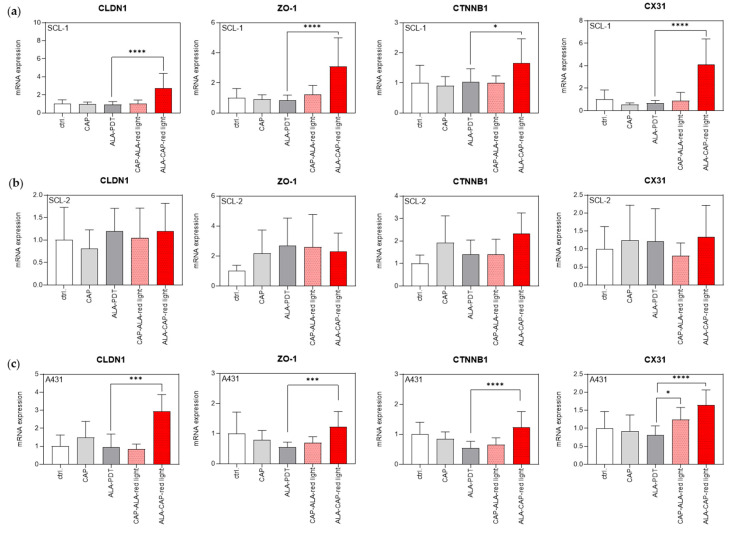
Expression of genes of the junctional network after different treatments (CAP, ALA-PDT, CAP-ALA–red light, or ALA-CAP–red light) and in untreated cells (ctrl.). (**a**–**c**) mRNA expression (CLDN1, ZO-1, CTNNB1, CX31) in cutaneous SCC cell lines (SCL-1, SCL-2, A431), (**d**–**f**) in oral SCC cell lines, and (**g**) in HEKn was measured 24 h after treatment. Statistical analysis: Ordinary one-way ANOVA with Bonferroni’s multiple comparison test was carried out to compare the mean of conventional ALA-PDT with combined treatments (CAP-ALA–red light, ALA-CAP–red light). * *p* ≤ 0.05, *** *p* < 0.001, **** *p* < 0.0001.

**Table 1 ijms-25-10808-t001:** Individual ALA concentration for SCC and HEKn cell culture experiments.

Cell Line	SCL-1	SCL-2	A431	SCC-9	SCC-15	SCC-111	HEKn
ALA [µM]	220	80	90	60	80	15	150

**Table 2 ijms-25-10808-t002:** Human primers and conditions.

Primer Name	Forward Primer 5′ → 3′	Reverse Primer 5′ → 3	Condition ^1^ (Annealing, Melting)
β-actin	CTACGTCGCCCTGGACTTCGAGC	GATGGAGCCGCCGATCCACACGG	ann. 60 °C, melt. 85 °C
Caspase-3	CTGCCGTGGTACAGAACTGG	TGGATGAACCAGGAGCCATC	ann. 60 °C, melt. 78 °C
Caspase-7	GACCGGTCCTCGTTTGTACC	TTCCGTTTCGAACGCCCATA	ann. 60 °C, melt. 78 °C
Caspase-9	TGAACTTCTGCCGTGAGTCC	CAGCAAAGCCAGCACCATTT	ann. 60 °C, melt. 84 °C
p16^INK4a^	GGAGCAGCATGGAGCCTTCGGC	CCACCAGCGTGTCCAGGAAGC	ann. 60 °C, melt. 89 °C
p21^CIP1^	CGAGGCACCGAGGCACTCAGAGG	CCTGCCTCCTCCCAACTCATCCC	ann. 60 °C, melt. 87 °C
ZO-1	GCCATTCCCGAAGGAGTTGA	GCAAAAGACCAACCGTCAGG	ann. 60 °C, melt. 81 °C
CX31	TCATCTTCGTCACATGCCCC	CTGCGTTGTCGTACAGCTTG	ann. 60 °C, melt. 87 °C
CLDN1	TTTACTCCTATGCCGGCGAC	GAGGATGCCAACCACCATCA	ann. 60 °C, melt. 84 °C
CTNNB1	TTTGATGGAGTTGGACATGGC	TGATGGTTCAGCCAAACGC	ann. 60 °C, melt. 81 °C

^1^ Quantitative real-time PCR was conducted with specific sets of primers and conditions. ann: annealing temperature; melt: melting temperature.

## Data Availability

Data are available upon reasonable request from the first and corresponding authors.

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
