# Peer review of "Optimization of the Treatment of Squamous Cell Carcinoma Cells by Combining Photodynamic Therapy with Cold Atmospheric Plasma"

_ijms, 2024, doi:10.3390/ijms251910808_

Round 1

Reviewer 1 Report (Previous Reviewer 1)

Comments and Suggestions for Authors

the authors did take my suggestions into account and the manuscript was improved

Author Response

Reviewer 2 Report (Previous Reviewer 2)

Comments and Suggestions for Authors

Summary: it is never explained how the process described could be used in a clinical setting. There are many other issues. A major limitation of PDT is penetration of light, not photosensitizer (line 18). What are ‘degenerated cells’ (line 45)? Reactive oxygen species indirectly cause cell death (line 71) by damaging host organelles, e.g., mitochondria. This can result in loss of cytochrome c into the cytoplasm, a trigger for apoptosis. Where is evidence that malignant cell types are more ‘sensitive’ to ROS than host cells (line 83)? Ref  12 refers to cold plasma selectivity. If the purpose of CAP is promoting ALA uptake, there are already analogs available with enhanced permeability features. 

Reliance on MTT data as an index of efficacy is a flawed approach. Oleinick showed (2009, Photochem. Photobiol.) That there is no correlation between MTT data and clonogenic results. MTT data are an unreliable index of efficacy. The term ‘cytotoxicity’ is essentially meaningless in the context of cancer therapy. If cells are not prevented from dividing, there can be no meaningful indication of efficacy. What does 20-30% cytotoxicity mean (lines 99-100)? Fig. 2 is incorrectly labeled ‘cell survival’ which is not being determined. Terms like ‘sublethal’ (line 115) are meaningless as is ‘underwent cytotoxicity’.

The Y axis of Fig. E is labeled ‘DCF’ but the legend claims that ROS are determined. Which is it? Figure legends need to be ‘self sufficient’. The legend tells the reader nothing about experimental conditions. Light doses (100 J/sq cm) are high for an in vitro studies (line 143). For some studies (Fig. 4), CAP alone appears to create more ROS than any other treatment. This indicates that addition of ALA + light actually reduces the effect. SCC-111 data show that ALA-PDT alone is essentially ineffective, but the sequence [ALA-CAP-red light] had a significant effect. This did not occur with other cell lines. The theory of ALA efficacy is that conversion to PpIX is minimal in normal host cells. But the ALA-PDT data show essentially similar ROS formation for all cell lines. The major difference occurs with SCC-111 cells. 

Migration and invasion are secondary to tumor eradication. Dead cells neither migrate nor invade. If cells are being ‘killed’, they would not be expected to do either. Apoptosis should occur promptly after mitochondrial photodamage or any other effect that should initiate the process. The point of waiting for 48 hours is never explained. It is periodically pointed out that a ‘very high’ ALA dose is used in HEKn cells. Why? The goal is showing effects on malignant vs. normal host cells at the same PDT doses, i.e., the same ALA doses. What occurs with regard to gene expression after PDT relates to cells that survive. The goal is tumor eradication. Legends to Figs. 7 and 8 do not indicate how long after treatment these data were acquired. 

In the Discussion, it is indicated that CAP promotes ALA permeabilty. There are ALA analogs already designed for this purpose. Why not use them? What does ‘approx. 25% of the cells underwent cytotoxicity (line 305) mean?  If the purpose of the study was to show that normal host cells were less affected by treatment than malignant cells, what was the point of using a high ALA dose with HEKn cells? 

In the Methodology section, it is indicated that CAP treatment involved treatment of cells under CPBS with what is described as ‘4 kHz, 2 min’ (line 441). The process is never explained, aside from a discussion (section 4.1) that this involves electrodes and a 3.5 kV voltage. How this would be used in vivo is never explained. Serum-free medium was sometimes used (line 447). This can initiate autophagy in some cell types and is never a good idea if in vitro studies are supposed to mimic what might occur in vivo. The light source (575-750) contains photons at many wavelengths that are not reflected in the PpIX absorbance spectrum, so light doses are not entirely relevant since they include irrelevant photons.    

Round 2

Reviewer 2 Report (Previous Reviewer 2)

Comments and Suggestions for Authors

In this revision, there is still considerable reliance on the MTT assay for assessing efficacy although Oleinick did show (2009) that MTT data were poorly correlated with clonogenic 

data which specifically relate to loss of viability. Calling MTT results ‘metabolic activity’ is not the answer. MTT data specifically reflect levels of mitochondrial dehydrogenases and is unrelated to photokilling, Why there is any reliance on such data is a mystery. If appropriate terms like ‘killing (line 92) are going to be used, then why even discuss MTT data? The authors  note (line 134) that the correlation is not there. There is no correlation but here are the data’.

Fig 2 is labeled ‘number of living cells’, but there is no information on how these data were obtained. What does ‘automated cell count’ mean (line 121)? If this is a growth assay, why present MTT data? It appears that different light/drug doses were sometimes used when effects on normal host cells vs. malignant cells were compared. This defeats the purpose of comparing  cell types under the same conditions. The term ‘metabolic activity’ appears again on line 381. 

The description of CAP (lines 86-88) indicates that is consists of ions, electrons, photons, reactive species and light. A gas cannot have light as a component since light consists of photons, not atoms or molecules. There may be photons present, but they cannot be called components of a gas. The CAP treatment method is described in section 4.1. A petri dish is apparently involved along with DPBS (line 510) since presence of serum causes problems (line 510-511). DPBS =  (line 498) Dulbecco’s phosphate buffered saline. Any in vivo studies will involve cells in equilibrium with blood plasma which is not DPBS. If serum quenches ROS, this will certainly occur in vivo when blood/serum is there. 

If use of CAP is a requirement for the proposed protocol, how this be used in vivo is never explained. Serum will necessarily be present. How will malignant tissues be exposed to CAP?  If Fig. 2b shows results of clonogenic assays, some cell lines do show significant efficacy of treatment, but the ALA concentration varied considerably for the different cell lines (Table 1), serum-free medium was used (a condition that can evoke an autophagic response) and it is not clear how the conditions used could be relevant to any in vivo situation. 

Round 3

Reviewer 2 Report (Previous Reviewer 2)

Comments and Suggestions for Authors

As indicated  in a prior review, it is never explained how the process described could be used in a clinical setting. The need for serum-free medium to show optimal efficacy is a significant cause for concern. How ‘cold plasma’ could be delivered in vivo is never explained. This may be a phenomenon that can only occur in vitro. There remains considerable reliance on MTT data which was shown to be irrelevant as an indicator of photokilling.

There continue to be phrases that make no sense. An example (lines 129-130): ‘The ALA concentration at which 70-80% metabolic activity is achieved . . . ‘. The authors point out (line 136) that MTT data are irrelevant so why report them? Fig. 2a contributes nothing to the argument. It should be explained in the legend to Fig. 2b how ‘number of living cells’ was determined. What does ‘CAP affects a broader area’ mean (line 178)? A broader area of what?

If the idea is to show that treatment is directed at neoplastic rather than normal host cells, it is necessary to use the same treatment parameters for both cell types. Varying these with the cell type only confuses this issue. The explanation for the use of varying light/drug doses on normal vs. malignant cells is not persuasive. The point of PDT is selective photodamage to malignant cells. Lines 276-278 illustrate the problems that can be created by varying the ALA concentration. How soon is apoptosis observed after treatment? It is necessary to wait for 48 hours? Many studies have shown that apoptotic cells are observed within 30-60 min after irradiation of photosensitizer cells. 

In the Discussion, it is claimed that the purpose of CAP is destabilization  of the ‘skin barrier’ to ALA uptake. Since there are ALA analogs with better permeability, it is not clear that such a treatment is needed. Moreover, it is never indicated how CAP would be used in a clinical setting. The need for serum-free medium suggests that this approach will be of possibly limited use where serum is present, e.g., in vivo. 

Every in vitro study does not need to be accompanied by an in vivo demonstration. But where there are indications that the presence of serum affects results, and where the protocol for CAP treatment in vivo is not obvious, it might be concluded that this approach will be irrelevant in a clinical setting. It appears to represent another cure for which there is no disease. What needs to be done: [1] indicate how CAP can be applied in vivo; [2] delete MTT data since they are unreliable for reporting on PDT efficacy; [3] use the same PDT dose for all studies since any in vivo application of PDT is going to involve a single PDT dose. 

Author Response

This manuscript is a resubmission of an earlier submission. The following is a list of the peer review reports and author responses from that submission.

Round 1

Reviewer 1 Report

Comments and Suggestions for Authors

the study was about the in-vitro assessmnet of the combined cold atmoscphric air and ALA-PDT to actinic keratosis. the study is well structured with:

1)a nice intro adding about the evolution of actinic keratosis and SCC 

Add more text that SCC can be de novo so would not make the misundertanding that actinic keratosis is an oblitaraory initial stage

2) results and methods excellent presented. nice tables and graphs

3)  brief report about clinical applications of your findings

4) write limitations of your study

5) write clinical scanarios the  combined cold atmoscphric air and ALA-PDT would be contra-indicated and why. 

Reviewer 2 Report

Comments and Suggestions for Authors

This report claims that the sequence of treatment procedures can affect the therapeutic response of malignant or pre-malignant cells to photodynamic therapy. There are several errors in the interpretation of results. The ‘MTT test’ tells us nothing about viability. In 2009, Oleinick published a report in Photohem. Photobiol. showing that the MTT results were not correlated with results obtained from clonogenic assays. Only clonogenic assays reliably report on ‘cell viability’. All that can be said from MTT data is that ‘cytotoxicity’ was observed: a poorly-defined term and not indicative of clonogenic results. 

In section 4.7, line 466 implies that DCFH-DA was present before irradiation, but this needs to be clearly indicated. In section 4.3, how was ‘lethality’ determined? If this involved the MTT assay, there are questions concerning the relevance of the data. Nothing from MTT data can be interpreted in terms of ‘viability’.  Since apoptosis was initiated after assorted treatments, it should not take 48 hours for this to become apparent (line 501). Since ALA is used as a pro-drug for PpIx formation, the relative permeability of tissues can be determinant of efficacy.

Fig. 2 can not be labeled ‘viability’ which can only be determined by clonogenic or other growth assays. Fig. 3 compares intrinsic levels of ROS in assorted cell types. Malignant cells tend to express a higher level than HEKn, but HEKn (human epidermal keratinocytes) are only one of many ‘normal’ cell types. There may be other such host cells that have intrinsically higher ROS levels, but in the context of skin cancer, this may not be relevant.  Fig. 2 discusses ‘activation’ of DCFH-DA when the author likely mean ‘oxidation’. It is not clear why a 48 hour interval was chosen to evaluate apoptosis. This should occur promptly after treatment. Apoptosis cells will be engulfed and eliminated by other host cells in vivo, but not in vitro.  

Effects of treatment of a collection of genes was examined. In the context of PDT, a typical progression of events involves [1] photodamage to mitochondria resulting in release of cytochrome c into the cytoplasm. [2] Interactions of cytochrome c leading to apoptosis. [3] Cell death. There may be changes in gene expression of dead and dying cells, but it is not clear whether this would significantly affect PDT efficacy.  It is not made clear how long after treatment effects on gene expressions were determined. Apoptosis will be initiated very soon after mitochondrial photodamage. 

Summary: effects on apoptosis need to be assessed soon after photodamage, not 48 hours after. Viability needs to be determined by clonogenic assays. If effects on gene expression require a substantial time to become detectable, an effective PDT dose will only lead to such changes occurring in dead and dying cells. If these changes are occurring in cells that can survive after photodamage, what is the relevance of these changes? A major flaw in the argument is the assumption that the MTT assay reflects anything but loss of mitochondrial dehydrogenase activity. No significant correlation with viability can be claimed.